
1    **Towards Objective Identification and Tracking of Convective Outflow Boundaries in Next-**

2    **Generation Geostationary Satellite Imagery**

Jason M. Apke[1], Kyle A. Hilburn[1], Steven D. Miller[1], and David A. Peterson[2]
[1]Cooperative Institute for Research in the Atmosphere (CIRA), Colorado State University, Fort
Collins, CO, USA
[2]Naval Research Laboratory, Monterey CA, USA

21   Corresponding Author:  Jason Apke

22   3925A West Laporte Ave. Fort Collins, CO 80523-1375

23   jason.apke@colostate.edu





## Abstract

Sudden wind direction and speed shifts from outflow boundaries (OFBs) associated with deep
convection significantly affect weather in the lower troposphere.  Specific OFB impacts include
rapid variation in wildfire spread rate and direction, the formation of convection, aviation hazards,
and degradation of visibility and air quality due to mineral dust aerosol lofting. Despite their
recognized importance to operational weather forecasters, OFB characterization (location, timing,
intensity, etc.) in numerical models remains challenging.  Thus, there remains a need for objective
OFB identification algorithms to assist decision support services.  With two operational next-
generation geostationary satellites now providing coverage over North America, high-temporal
and spatial resolution satellite imagery provides a unique resource for OFB identification.  A
system is conceptualized here designed around the new capabilities to objectively derive dense
mesoscale motion flow fields in the Geostationary Operational Environmental Satellite (GOES)-
16 imagery via optical flow.  OFBs are identified here by isolating linear features in satellite
imagery, and back-tracking them using optical flow to determine if they originated from a deep
convection source.  This "objective OFB identification" is tested with a case study of an OFB
triggered dust storm over southern Arizona.  Results highlight the importance of motion
discontinuity preservation, revealing that standard optical flow algorithms used with previous
studies underestimate wind speeds when background pixels are included in the computation with
cloud targets.  The primary source of false alarms is incorrect identification of line-like features in
the initial satellite imagery.  Future improvements to this process are described to ultimately
provide a fully automated OFB identification algorithm.



## 1. Introduction

Downburst outflows from associated deep convection (Byers and Braham Jr., 1949; Mitchell and Hovermale, 1977) play a significant, dynamic role in modulation of the lower troposphere. Their direct impacts to society are readily apparent—capsizing boats on lakes and rivers with winds that seem to "*come out of nowhere*" (e.g. The Branson, MO duck boat accident; Associated Press 2018), causing shifts in wildfire motion and fire intensity that put firefighters in harm's way (e.g. the Waldo Canyon and Yarnell Hill Fires; Hardy and Comfort, 2015; Johnson et al., 2014), and threatening aviation safety at regional airports with sudden shifts from head to tail-winds and turbulent wakes (Klingle et al., 1987; Uyeda and Zrnić, 1986). In the desert southwest, convective outflows can loft immense amounts of dust, significantly reducing surface visibility and air quality for those within the impacted area (e.g. Idso et al. 1972; Raman et al. 2014). These outflows are commonly associated with rapid temperature, pressure, and moisture changes at the surface (Mahoney III, 1988). Furthermore, the collision of outflows from adjacent storms can serve as the focal point of incipient convection or the intensification of nascent storms (Rotunno et al., 1988; Mueller et al., 2003).

Despite the understood importance of deep convection and convectively driven outflows, high resolution models struggle to characterize and identify them (e.g. Yin et al. 2005). At present, outflow boundaries (OFBs) are instead most effectively monitored in real-time at operational centers around the world with surface, radar, and satellite data. Satellites often offer the only form of observation in remote locations. The most common method for detecting outflows via satellite data involves the identification of clouds formed by strong convergence at the OFB leading edge. When the lower troposphere is dry, OFBs may be demarcated by an airborne "dust front", after passing over certain surfaces prone to deflation by frictional winds (Miller et al., 2008). The task



of identifying OFBs can prove quite challenging and would benefit greatly from an objective
means of feature identification and tracking for better decision support services.
The Advanced Baseline Imager (ABI), an imaging radiometer carried on board the
Geostationary Operational Environmental Satellite (GOES)-R era systems, offers a leap forward
in capabilities for the real-time monitoring and characterization of OFBs. Its markedly improved
spatial (0.5 vs. 1.0 km visible, 2 km vs. 4 km infrared), spectral (16 vs. 5 spectral bands), and
temporal (5 min vs. 30 min continental U.S., and 15 min vs. 3 hr full disk) resolution provides new
opportunities for passive sampling of the atmosphere over the previous generation (Schmit et al.,
2016). The vast improvement of temporal resolution alone (which includes mesoscale sectors that
refresh as high as 30 s) allows for dramatically improved tracking of convection (Cintineo et al.,
2014; Mecikalski et al., 2016; Sieglaff et al., 2013), fires and pyroconvection (Peterson et al., 2015,
2017, 2018), ice flows, and synoptic scale patterns (Line et al., 2016). This higher temporal
resolution makes identification of features like OFBs easier as well because of greater frame-to-
frame consistency.
The goal of this work is to use ABI information towards objective identification of OFBs. One
of the notable challenges in satellite identification of OFBs over radar or models is the lack of
auxiliary information. When working with a radar or a numerical model framework, for example,
additional information is available on the flow, temperature, and pressure tendency of the
boundary. Without that information, however, forecasters must rely on their knowledge of gust
front dynamics to identify OFBs in satellite imagery. Here, we introduce the concept of objectively
derived motion using GOES-16 ABI imagery for feature identification via an advanced optical
flow method, customized to the problem at hand. A case study of a convectively triggered OFB
and accompanying haboob dust front is presented in 5-min GOES-16 contiguous United States



(CONUS) sector information, as a way of evaluating and illustrating the potential of the
framework.

This paper is outlined as follows. The background for objective motion extraction and OFB

identification is presented in Section 2. The optical flow methods developed for this purpose are
discussed in Section 3. Section 4 presents the case study test of the current algorithm, and Section
5 concludes the paper with a discussion on plans for future work in objective feature identification
from next-generation geostationary imagers of similar fidelity to the GOES-R ABI, which are
presently coming online around the globe.

## 105    2.    Background

*2.1 Previous Work in OFB Detection*

Objective identification of OFBs in meteorological data has been a topic of scientific inquiry

for more than 30 years. Uyeda and Zrnić (1986) and Hermes et al. (1993) use detections of wind
shifts in terminal Doppler radar velocity measurements to isolate regions of strong radial shear
associated with OFBs. Smalley et al. (2007) include the "fine line" reflectivity structure of
biological- and precipitation-sized particles to identify OFBs via image template matching.
Chipilski et al. (2018) considered the OFB objective identification in numerical models using
similar image processing techniques, but with additional dynamical constraints on vertical velocity
magnitudes and mean-sea level pressure tendency. Objective OFB identification has not been
demonstrated to date with the new ABI observations of the GOES-R satellite series. Identification
via satellite imagery would be valuable for local deep convection nowcasting algorithms which
use boundary presence as a predictor field (Mueller et al., 2003; Roberts et al., 2012), and for



operational centers around the world which may not have access to ground-based Doppler radar
data.

Traditionally, forecasters have identified OFBs in satellite imagery by visually identifying

quasi-linear low-level cloud features and back-tracking them to an associated deep convection
source. Previous objective motion derivation algorithms are not designed to yield the dense wind
fields necessary for identifying and tracking features such as OFBs (Bedka et al., 2009; Velden et
al., 2005). In fact, the original image window-matching atmospheric motion vector (AMV)
algorithms produce winds only over targets deemed acceptable for tracking by pre-processing
checks on the number of cloud layers in a scene, brightness gradient strength, and patch coherency.
The targets are further filtered with post-processing checks on acceleration and curvature through
three-frame motion and deviation from numerical model flow (Bresky et al., 2012; Nieman et al.,
1997; Velden et al., 1997). These practices were followed for a very practical reason—AMV
algorithms were tailored for model data assimilation. In the formation of the model analysis,
observational data must be heavily quality-controlled with outliers removed, to minimize data
rejection. Here, information such as OFBs would be rejected due to the detailed space/time
structure of actual convection which is typically poorly represented by the numerical model.

Deriving two-dimensional flow information at every point in the imagery would require either

modification of previous AMV schemes or post-processing of the AMV data via objective analysis
(e.g. Apke et al. 2018). The latter typically will not capture motion field discontinuities, resulting
in incorrect flows near feature edges (Apke et al., 2016). To capture such discontinuities in a dense
flow algorithm, new computer vision techniques, such as the gradient-based methods of optical
flow, must be adopted.



### 2.2 Optical Flow Techniques

Optical flow gradient-based techniques derive motion within fixed windows, thus eliminating the reliance on models for defining a search region. A core assumption of many optical flow techniques is brightness constancy (Horn and Schunck, 1981). Considering two image frames, brightness constancy states that the image intensity $I$ at some point $\boldsymbol{x} = [x, y]^T$ is equal to the image intensity in the subsequent frame at a new point, $\boldsymbol{x} + \boldsymbol{U}$, where $\boldsymbol{U} = [u, v]^T$ represents the flow components of the image over the time interval ($\Delta t$) between the two images:

$$I(\boldsymbol{x}, t) = I(\boldsymbol{x} + \boldsymbol{U}, t + \Delta t) \tag{1}$$

Eq. (1) can be linearized to solve for the individual flow components, $u$ and $v$:

$$\nabla I \cdot \boldsymbol{U} + I_t = 0 \tag{2}$$

Where $\nabla I = [I_x, \ I_y]$ represents the intensity gradients in the x and y direction, and $I_t$ represents the temporal gradient of intensity. For one image pixel, Eq. (2) contains two unknowns with a simple translation model for $\boldsymbol{U}$; therefore, it cannot be solved point-wise. One well-known approach to solving this so-called "aperture problem" is the Lucas-Kanade, hereafter LK method, which considers a measurement neighborhood of the intensity space and time gradients (e.g., Baker and Matthews, 2004; Bresky and Daniels, 2006). Use of neighborhoods, or image windows, to derive optical flow are called *local* approaches. Another seminal approach was introduced by Horn and Schunck (1981; HS Method) which solves the aperture problem by adding an additional smoothness constraint to the brightness constancy assumption, and minimizing an energy magnitude between two images:

$$E(u, v) = \iint\limits_{\Omega} (\nabla I \cdot \boldsymbol{U} + I_t)^2 + \alpha(|\nabla_2 u|^2 + |\nabla_2 v|^2) \ d\mathbf{x} \tag{3}$$





Where $E(u,v)$ represents an energy functional to be minimized over all image pixels $\Omega$, $\alpha$ is a
constant weight used to control the smoothness of the flow components $u(\mathbf{x})$ and $v(\mathbf{x})$, and $\nabla_2=$
$[\partial/\partial\mathrm{x}, \partial/\partial\mathrm{y}]^T$. This optical flow derivation is called a *global* approach. Eq. (3) is minimized in
Horn and Schunck (1981) by deriving the Euler-Lagrange equations, and numerically solving with
Gauss-Seidel iterations.

Linearizing the brightness constancy equation into Eq. (2) means that large and non-linear

displacements (typically > 1 pixel between images) will not be captured (Brox et al., 2004). Thus,
most optical flow computations initially spatially subsample images to where all displacements
are initially less than 1-pixel (Anandan, 1989; discussed more in Section 3.1), which can cause
fast moving small features to be lost. Note that reducing the temporal resolution of GOES imagery
(e.g. 10-min vs. 5-min scans) increases the displacement of typical meteorological features
between frames. Furthermore, constancy assumptions are more likely violated with reduced
temporal resolution since image intensity changes more through evaporation and condensation of
cloud matter over time. Thus, for the spatial resolution of ABI, it is impractical to consider optical
flow gradient-based methods at temporal resolutions coarser than 5-min for several mesoscale
meteorological phenomena, including OFBs. Very spatially coarse images do not need to be
initially used with faster scanning rates, such as super rapid scan 1-min information (Schmit et al.,
2013), or the 30-s temporal resolution mesoscale mode of ABI (Schmit et al., 2016).

While both the LK and HS methods are designed for deriving dense flow in satellite imagery,

neither account for motion discontinuities in fields. Hence, both suffer from incorrect flow
derivations near cloud edges, and would perform poorly for OFB detection and tracking. Black
and Anandan (1996) offer an intuitive solution to this problem, whereby the energy functional is
designed to minimize robust functions that are not sensitive to outliers:



$$E(u,v) = \iint_\Omega \rho_d(\nabla I \cdot \boldsymbol{U} + I_t) + \rho_s(|\nabla_2 u|^2 + |\nabla_2 v|^2)d\mathbf{x} \tag{4}$$

The robust function data term for the standard HS approach is simply $\rho_d(r) = r^2$, and smoothness
$\rho_s(r) = r$ which implies that energy functionals increase quadratically for $r$ outliers. Other robust
functions can also be minimized with similar gradient descent algorithms to Gauss-Seidel
iterations, while being less sensitive to outliers (Press et al., 1992; Black and Anandan, 1996).
Robust functions are popular in recent optical flow work (Brox et al., 2004; Sun et al., 2010), and
a similar approach was adopted here and is discussed further in the methodology section. The
reader is referred to works by Barron et al. (1994), Fleet and Weiss (2005), and Sun et al. (2010)
for a more comprehensive review on optical flow techniques.
The relevance of optical flow in satellite meteorological research continues to increase now
that scanning rates of sensors such as the ABI are routinely at sub 5-min time scales, making
motion easier to derive objectively (Bresky and Daniels, 2006; Héas et al., 2007; Wu et al., 2016).
The dense motion estimation within fine-temporal resolution data has yet to be used for feature
identification. Optimizing optical flow for this purpose, and its specific application to OFBs, is the
aim of this study. The next section outlines our approach to this end.

**3.  Methodology**
*3.1 Optical Flow Approach*
A recent approach to handle piecewise and non-linear image changes in flow is introduced by
Brox et al. (2004) (Hereafter B04), where the brightness constancy assumption is no longer
linearized, i.e.





$$E(u,v) = \iint_{\Omega} \rho_d(|I(\boldsymbol{x} + \boldsymbol{U}, t + \Delta t) - I(\boldsymbol{x}, t)|^2$$

$$+ \gamma |\nabla_2 I(\boldsymbol{x} + \boldsymbol{U}, t + \Delta t) - \nabla_2 I(\boldsymbol{x}, t)|^2)$$

$$+ \alpha\, \rho_s(|\nabla_2 u|^2 + |\nabla_2 v|^2) d\boldsymbol{x} \tag{5}$$

Following B04, within the data robust function, we now have also included a gradient constancy
assumption, which is weighted by a constant $\gamma$ to make the derived flow more resilient to changes
in illumination. Avoiding linearization of constancy assumptions improves the identification of
large displacements between images. The Charbonnier penalty is used for the data and smoothness
robust functions following Sun et al. (2010),

$$\rho_d(r^2) = \rho_s(r^2) = \sqrt{r^2 + \epsilon^2} \tag{6}$$

with $\epsilon$ representing a small constant present to prevent division by zero in minimization, set to
0.001. The values for $\boldsymbol{U}$ are found by solving the Euler-Lagrange equations of Eq. (5) with
numerical methods

$$E_u - \frac{dE_{u_x}}{dx} - \frac{dE_{u_y}}{dy} = 0 \tag{7}$$

$$E_v - \frac{dE_{v_x}}{dx} - \frac{dE_{v_y}}{dy} = 0 \tag{8}$$

with reflecting boundary conditions and subscripts that imply the derivatives. Eqs. (7) and (8) are
solved with a nested-fixed point successive over-relaxation iteration scheme described in B04 and
summarized in Fig. 1. The reader is referred to Chapter 4 of Brox (2005) for details on the full
discretization of the derivatives in the successive over-relaxation scheme. Here, only the spatial
dimensions are used for the smoothing term, though it is possible to include the time dimension
with this system as well.





A difficulty in solving Eqs. (7) and (8) is that the successive over-relaxation scheme may
converge on a local minimum of $E(u, v)$, rather than finding the global minimum.  The typical
approach to find the global minimum is to compute optical flow with coarse- to fine-scale warping
iterations (e.g. Anandan, 1989).  Coarse-to fine-scale warping iterations work by subsampling the
initial image at native resolution to a coarser spatial resolution and computing the flow initially at
the coarsest resolution in the image pyramid.  The $U$ results from the coarse image flow are then
used as the first guess field to for the next finest scale on the image pyramid (Fig. 2), and the
second image is warped accordingly.  The warping step ensures that estimated displacements at
every step in the image pyramid remain small.
The B04 scheme includes coarse-to fine-scale warping iterations at every outer iteration $k$.
This means that the first iteration is run on a subsampled image, and the subsampling is reduced
by a scale factor at every $k$ until the image reaches native resolution at the final $k = nK$.  Images
at every $k$ in this subsampling are found using a gaussian image pyramid technique with bicubic
interpolation.  The flow values of the image at $k - 1$ are upscaled accordingly at $k$ also with
bicubic interpolation (the initial flow guess is $u = v = 0$ at $k = 0$).  For improved computation of
spatial derivatives, the initial image is also smoothed with a 9x9 pixel kernel gaussian filter with a
standard deviation set to 1.5 pixels.  The specific settings used for the coarse- to fine- warped flow
scheme here are shown in Table 1.
*3.2 Objective OFB identification*
There are two steps to the objective OFB identification process.  First, a linear feature or sharp
boundary is identified in visible or infrared imagery.  In some cases, the first step alone is enough
to identify OFBs subjectively.  The second step is tracking that feature back in time to see where
it originated from (typically, near an area with deep convection).  In the case of near stationary





convection and low-level flow, a forecaster might also use radial like propagation in this decision-
making process, however, since convection geometry and low-level flow varies from storm to
storm, only the first two steps are considered here. This approach aims to mirror the subjective
process, leveraging the information content of optical flow to do so.

To handle the first step of line feature identification, a simple image line detection scheme was

performed using the sum of a set of two-dimensional convolution kernels:
$$a_1 = \begin{bmatrix} -1 & -1 & -1 \\ 2 & 2 & 2 \\ -1 & -1 & -1 \end{bmatrix} a_2 = \begin{bmatrix} -1 & 2 & -1 \\ -1 & 2 & -1 \\ -1 & 2 & -1 \end{bmatrix} a_3 = \begin{bmatrix} 2 & -1 & -1 \\ -1 & 2 & -1 \\ -1 & -1 & 2 \end{bmatrix} a_4 = \begin{bmatrix} -1 & -1 & 2 \\ -1 & 2 & -1 \\ 2 & -1 & -1 \end{bmatrix}$$

Applying these kernels to the gaussian smoothed ABI visible imagery (using a 21x21 kernel and
standard deviation of 5 pixels) results in high intensity values where line structures exist. A two-
dimensional convolution threshold of 0.02 was used with the visible imagery calibrated to
reflectance factor to isolate line features. This method was compared to a subjective interpretation
of boundary location for validation.

To address the second step of the process, the constrained optical flow approach described in

Section 3.1 was used to track the boundary pixels (both objectively and subjectively identified)
back in time for three hours. The values of motion at each step in the backwards trajectory were
determined with bilinear interpolation of the optical flow derived dense vector grid. If a back-
traced pixel of the linear feature arrived within 50 km of a convective area with a 10.35 μm
brightness temperature ($BT_{10.35}$) lower than 223 K (-50 °C; using previous satellite imagery
matched to the back-trajectory time), the original point was considered an OFB. While this
brightness temperature threshold is subjective and can vary from case to case, it was found to
produce a reasonable approximation of deep convection areas when compared to ground-based
radar information for the case study described in the subsequent sections.





*3.3 Data*

The objective OFB identification methodology is tested using a case study from 5 July 2018

over the southwestern United States. This event featured a distinct OFB and associated dust storm
that was well-sampled by various ground- and space- based sensors. GOES-16 was in Mode-3,
generating one image over the study area every 5-min (continental U.S., or CONUS, ABI scan
domain). Optical flow computations employ the GOES-16 (GOES-East) ABI red band (0.64 μm;
ABI channel 2), provided at a nominal sub-satellite spatial resolution of 500 m, but closer to 1 km
at the case study location. This channel is used at native resolution, though it can be subsampled
with a low-pass filter such that future versions can implement color information from the blue and
near-infrared bands (e.g. Miller et al. 2012). This means that the optical flow approach here is
daytime only. A similar B04 approach can be used on infrared data as well for day/night
independent information, though for detecting OFBs in the low levels, proxy visible products
would perform best. As described above, the clean longwave infrared band (10.35 μm; ABI
channel 13) is used as first-order information on optically thick cloud-top heights and to assess the
convective nature of the observed scene ($BT_{10.35} < 223$ K).

High frequency Automated Surface Observing Stations (ASOS; NOAA 1998), recording

temperature, pressure, wind speed and direction once every minute, complement the satellite
imagery. The Weather Surveillance Radar-1988 Doppler (Crum and Alberty, 1993) dual-
polarimetric data also sampled the OFB event from the KIWA radar near Phoenix, AZ. To
highlight the OFBs and the presence of dust, horizontal reflectivity and correlation coefficient are
used (Van Den Broeke and Alsarraf, 2016). Finally, for information on the full 3D dynamics of
the case study, a numerical model representation of the environment was collected from the High
Resolution Rapid Refresh system (HRRR, Benjamin et al. 2016). The combination of these model



and observation datasets is employed to confirm the presence of a distinct convective OFB, rather
than some other quasi-linear feature, such as a bore or elevated cloud layer, etc.

**4.  Case Study Description**

Convection was observed in south central Arizona on 5 July 2018 after 1800 UTC.  A large
and well-defined linear structure emerged from below the convective cloud cover at 2200 UTC to
6 July 2018 0100 UTC propagating westward in GOES-16 imagery (Fig. 3).  This linear structure,
demarcated by roll (arcus) clouds on the northern side and lofted dust on the southern side, was
apparent with strong visible reflectance contrast against the relatively dark surface and $BT_{10.35}$ ~
10 K cooler than the underlying surface.  The dust lofted by this outflow produced low visibility
and hazardous driving conditions near Phoenix, AZ.  Dust storm warnings were issued by the local
National Weather Service (NWS) forecast office by 2300 UTC.  The structure's observed radial
propagation away from nearby deep convection and associated cloud and dust features lends to its
interpretation as a convective OFB.
The OFB was also captured in radar scans from KIWA at 2200 UTC (Fig. 4).  The coincidence
of low correlation coefficient ($< \sim 0.5$) and moderate to high reflectivity (near 20 dBZ) imply that
the OFB was associated with airborne coarse dust particles.  Surface observations taken at the
ASOS station reveal temperatures exceeding 317 K (44 °C) ahead of the OFB, with calm winds
(Fig. 5).  Temperatures dropped by 4 K, wind speeds changed direction and increased sharply, and
dew points increased rapidly as the OFB crossed the station at ~2316 UTC.  The rapid change in
low-level meteorology is consistent with convective OFBs sampled in previous studies (e.g.
Mahoney III, 1988; Miller et al., 2008).





The HRRR model captured the broad characteristics of this event (Fig. 6), showing moderate
low-level winds in excess of 10 m s$^{-1}$ (Fig. 6a), cooler temperatures (Fig. 6b), and simulated
cumulus clouds from forced ascent (Fig. 6c).  Model cross sections (Fig. 6d) indicated a moderate
increase in vertical motion ahead of the numerically derived boundary, and a sharp decrease in
virtual potential temperature behind the boundary.  The shape of the virtual potential temperature
profile is consistent with other model observations of OFBs (e.g. Chipilski et al., 2018).  The
observation and model data all show that the linear structure observed in Fig. 3 was modifying the
dynamics of the surface in a manner consistent with OFBs, and not some other linear cloud feature
type that is decoupled from the surface and may be misidentified by the satellite.  Since such low-
level linear features are often obscured by cloud layers at higher altitudes, this case study in some
respects represents a best-case-scenario for evaluating optical flow capabilities towards identifying
OFBs.

**5.   Results**
The first step in OFB identification requires identification of a feature that appears linear in
the imagery.  Compared to the subjective boundary identification (considered as truth here; Fig 7a,
blue dots), the convolution method gives a reasonable approximation to where the OFB is located
within the higher intensity points in the convolution (Fig. 7b).  Unfortunately, the simply-applied
convolution is also sensitive to linear features associated with the deep convection itself (the blue
shading in Fig. 7b).  Hence, false alarms appear east of the boundary.  These issues can be filtered
out using either cloud-top height or brightness temperature thresholding from separate infrared
channels.  Alternatively, the storm-relative motion (here > 15 m s$^{-1}$) from optical flow was used
here to filter the false alarms (the red shading in Fig. 7b).



The second step requires these linear fast-moving features to be traced backward to a deep
convection source using the optical flow computation (Fig. 8). To the west of the boundary, near
stationary optical flow vectors highlight the background (or ground) pixels. The boundary itself
exhibits a westward movement near 15-20 m s$^{-1}$ (~30-40 kts). The feature also appears to bow
outwards after faster motions are observed, near 33° N, -112° E during 2338-2358 UTC (Figs. 8b,
c). Similar westward motion is derived in the wake of the OFB, within the convective cold pool.
This results from the presence of airborne dust particles, which facilitate the computation of optical
flow vectors in this region.
The backwards trajectories of the subjectively and objectively identified OFB pixels in Figs.
7c and d (B04 method) show that many of the linear cloud features, particularly those associated
with the central arcus cloud, indeed originated near deep convection. However, when the
backwards trajectories of the B04 method were compared to other optical flow methods, such as
the approach by Wu et al. (2016), most were unsuccessful at obtaining coincidence between linear
cloud features along the OFB and a deep convection source. Wu et al. (2016) used an approach
introduced to the community by Farnebäck (2001), which is a *local* window method for optical
flow.
Example points 1–7 examined within the subjectively identified OFB backward trajectories
highlight an issue with *local* window approaches for this application (Fig. 9). The B04 approach
(Fig. 9, blue/yellow) produced motions that were relatively consistent with the true boundary
motion. Thus, many points that are lost in the *local* approaches are successfully backtracked to
the initial deep convection (e.g. points 3–5). With the Wu et al. approach (Fig. 9, orange/red),
OFB targets move slower than the actual boundary, and, over a three-hour tracking period,
eventually become stuck within the stationary background pixels. This tracking issue stems from



an assumption made in many *local* approaches that pixels within an image window all move in the
same direction with the same speed.  When background pixels are included within an image
window containing clouds or dust, the resulting optical flow speed would then be underestimated.
The slow bias is observed in plots of optical flow speeds along the OFB (Fig. 10), where the Wu
et al. approach was ~5-10 m s$^{-1}$ slower than the B04 approach.  While not shown, we found similar
backward trajectory issues using the LK approach.  Full loops of the optical flow in Fig. 8 and
trajectories in Fig. 9 are included as supplementary material to this manuscript.

For all approaches tested, however, methods struggled to backtrack the newly formed cumulus

to the north and the dust front to the south.  With the cumulus to the north, the issues with each
algorithm appear to result from rapid cumulus development between frames (e.g. points 1 and 2 in
Figs. 9a, b).  Condensation like what is observed here is unfortunately not considered in the
brightness constancy assumption.  Thus, condensing cloud features would only be tracked back to
when they initially form (after Fig. 9b) without additional dynamic constraints to Eq. (5).  An
example can be seen when points 1 and 2 become stuck in Fig. 9c.  This has important implications
on limitations of backtracking OFB features to deep convection with optical flow from imagery.
If no cloud or dust feature exists to visualize an OFB in satellite imagery, some of the feature
propagation may be lost.

The dust to the south appears in the satellite imagery as early as 2200 UTC, though it was quite

transparent relative to the ground.  It is therefore possible the stationary background pixels may be
dominant in the optical flow computation at points 6 and 7, resulting in slower wind speeds than
the true OFB propagation.  Points 6 and 7 are also located near cumulus moving across the OFB
motion to the south.  This dust front tracking could be improved using multispectral techniques
designed to highlight dust features over ground pixels, or by using additional color spectrum



information to discourage flow smoothness in Eq. (5) across the dust front from the cumulus to

the south (e.g. Sun et al., 2010).

Many line-like targets east of the OFB in Fig. 7d also originated from the deep convection,

which constitute false alarms. These false alarms can be reduced by further improving the OFB

targeting step in the objective process in future studies. For this case study, it may have been

possible to use convergence methods, analogous to radar-based objective OFB identification, to

isolate the boundary. However, convergence as derived from the optical flow information here

would only work because of local, stationary background pixels ahead of the OFB. This means

that convergence would almost always be inferred from OFB motion when the background is

involved. This approach would also be sensitive to nearby cloud structures ahead of the OFB

which would exhibit different (non-stationary) motion from the background. It is for this reason

that a backwards trajectory approach was elected instead of basing the detection on local horizontal

convergence. The optical flow approach used here does help highlight the OFB when storm

motion alone was considered in addition to convolution, showing how additional tools can be used

in synergy to arrive at a more comprehensive objective feature identification approach in future

studies.

## 6.  Conclusions and Future Outlook

A new method for the objective identification of outflow boundaries (OFBs) in GOES-16

Advanced Baseline Imager (ABI) data was developed using optical flow motion derivation

algorithms and demonstrated with provisional success on a dust storm case study. An optical flow

system constructed for this purpose shows promise in identifying and backtracking object events

to their source over traditional flow derivation methods, which can potentially be used to isolate





convective OFB features. To the best of the authors' knowledge, this study represents a first
attempt to objectively identify OFBs in geostationary satellite imagery.
The primary conclusions of this study are that optical flow approaches are now a viable option
to acquire meso-scale flows relevant to OFB tracking and detection in 5-min geostationary satellite
imagery, though the successful backtracking of OFB features requires use of flow algorithms that
can handle the presence of motion discontinuities and stationary background flow. The optical
flow algorithm tested in this study produced a dense motion field that was closer than other
methods to the true OFB motion and provided valuable information towards full objective OFB
identification in new products.
While several OFB related image pixels were successfully identified, the algorithm here is
relatively immature and remains fraught with false alarms where linear features are incorrectly
identified, and where correct features were not successfully backtracked to deep convection. The
algorithm is still limited by the assumptions made within optical flow, which only account for
changes in image brightness intensity resulting from pure feature advection. Therefore, if no
features (e.g. clouds) exist to highlight an OFB boundary within the imagery, the method proposed
here would not function properly. The method also struggles to resolve true OFB motions with
transparent dust movement, where a textured background beneath the dust may dominate the
motion estimate within a scene. Also, while infrared brightness temperature was enough to
identify deep convection in this case study, convection may be missed by brightness temperature
imagery if it is obscured by a higher cloud layer, or if the minimum cloud-top brightness
temperature exceeds an arbitrarily set threshold.
Given these limitations, future studies will explore more advanced systems for linear structure
identification to identify candidate features for tracking towards full objective OFB identification.



A machine learning system will be used to determine which linear characteristics of the image
should be backtracked instead of using two-dimensional convolution. Optical flow can be used to
precondition training information for a machine learning approach, if motion or semi-Lagrangian
fields are needed. Furthermore, it will be prudent to use deep convection correspondence through
optical flow backtracking as one of many fields in future products, such as radial propagation away
from storms and near surface meteorological properties, to probabilistically decide if an image
pixel is associated with an OFB. To better identify deep convection areas, the GOES Lightning
Mapper (GLM) can be used, which provides information on lightning location and energy at 8 km
resolution with a 2 ms frame rate.
Feature identification with optical flow is not restricted to OFBs alone. For example, the
above-anvil cirrus plume (Bedka et al., 2018) over deep convection has been identified as an
important indicator of severe weather at the ground, yet no objective means of identification exists
today. The properties from optical flow could be used as an additional source of information in
such algorithm designs, allowing researchers to backtrack features to their apparent source (the
overshooting top in the case of the above-anvil cirrus plume) and monitor cloud temperature and
visible texture trends, or to simply use the dense motion itself to achieve better results. This
method will also be applicable to other cold pool outflow phenomena, such as bores, for which
new algorithms could utilize numerical model or surface observations for further clarification of
linear feature type.
Motion discontinuity preserving optical flow will also benefit several current algorithms for
monitoring deep convection in satellite imagery. Objective deep convection cloud-top flow field
algorithms (Apke et al., 2016, 2018) will benefit particularly when sharp cloud-edges and ground
pixels are present in an image scene. Systems that use infrared cloud-top cooling or emissivity





differences for deep convection nowcasting will also improve with better estimates of pre-
convective cumulus motion (Cintineo et al., 2014; Mecikalski and Bedka, 2006).

While the utility of a backwards trajectory approach was considered here, many other possible

methods exist for exploiting the semi-Lagrangian properties of time-resolved observations in
satellite imagery (e.g. Nisi et al., 2014). Use of fine-temporal resolution information will improve
optical flow estimates, and in turn the estimates of brightness temperature, reflectance, or cloud-
property changes in a moving frame of reference. We will explore these and other refinements in
ongoing and future work on this exciting frontier of next-generation ABI-enabled science.

**7.   Data Availability**

Data used in this study are available upon request by contacting the lead author.


**8.   Author Contributions**

JMA developed the primary code for the optical flow approach used here. He also co-

developed the objective outflow boundary identification techniques, and related Figs. (1, 2, 3, 7-
10) in the manuscript. He co-wrote much of the text and led the efforts of interpretation, analysis,
and presentation of the results.

KAH was responsible for case study identification, and collection of surface, radar, and HRRR

data relevant to this case study. He developed Figs. 4, 5 and 6. He also co-developed the objective
OFB identification process and co-wrote the text.

SDM was the PI of the Multidisciplinary University Research Initiative (MURI) research

project and was responsible for managing the development of the optical flow code and outflow
boundary case study information involved in Figs. (1, 2, 3, 7-10). He co-developed the objective



identification process and maintained and managed the satellite data necessary to complete the
study.  He also co-wrote much of the text within the manuscript.

DAP co-developed the objective OFB identification scheme.  He also co-developed the

conceptual Figs. 1 and 2 to add clarity to the optical flow process used here and co-wrote the text
within the manuscript.

**9.    Competing Interests**

The authors declare that they have no conflicts of interest.


**10.   Acknowledgements**
This CIRA team was funded by the Multidisciplinary University Research Initiative (MURI) grant
N00014-16-1-2040.   David Peterson was supported by the National Aeronautics and Space
Administration (NASA) award NNH17ZDA001N.  Our special thanks to Dan Bikos and Curtis
Seaman at the Cooperative Institute for Research in the Atmosphere for informative discussions
on identification of outflow boundaries in satellite imagery.  We also thank Max Marchand for
providing the high-frequency surface observations used in this study.



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

**Table 1.** Settings used in the Brox et al. (2004) successive over-relaxation scheme.





13. **List of Figures**

**Figure 1.** Flow chart of the B04 optical flow approach used here. Note that SF, nK, nL and nM

are defined in Table 1.

**Figure 2.** Schematic of coarse- to fine-scale warping optical flow in GOES-imagery. The largest

displacements are found in the initial coarse grid (yellow arrow at the top of the pyramid), which

are used as initial displacements for the next levels (red and blue arrows). The final

displacement is the sum of each displacement estimate (white arrow). In this schematic, an

example scale factor of 0.5 was used over 3 pyramid levels, in this work, a scale factor of 0.95

for 77 levels was used.

**Figure 3.** The 6 July 2018 0023 UTC GOES-16 0.64-μm visible reflectance (top) and $BT_{10.35}$

(bottom) over south-central AZ, centered on an OFB of interest.

**Figure 4.** The KIWA Radar 2244 UTC 0.5° horizontal reflectivity (top) in dBZ and correlation

coefficient (bottom). Range rings in grey indicate every 30° azimuth and 50 km in range.

**Figure 5.** Surface High Frequency METAR observations of temperature (K; top left), dewpoint

(K; top right), mean sea level pressure (middle left), wind direction (° from N; middle right),

wind speed (m s$^{-1}$; bottom left), and wind gusts (m s$^{-1}$; bottom right). The surface station was

located at (32.95 °N -111.77 °E). The red line indicates the approximate time of boundary

passage over the station.

**Figure 6.** Four panel of HRRR output of OFB event, including a) wind speed, b) temperature, c)

simulated infrared brightness temperature, and d) a cross section along the black line in c with

virtual potential temperature $\theta v$ in black contours (K), omega in color shaded pixels, and regions

of relative humidity > 90% highlighted with dark shading (bottom right).





**Figure 7.** The 0023 UTC GOES-16 0.64-μm visible channel shown with a) subjectively

identified OFB (blue dots) and b) objectively identified linear features (blue shading). Also

shown are linear features that contained fast storm-relative motion (red shading). The results of

backtracking the c) subjectively and d) objectively identified OFB features are also shown,

where blue dots represent targets tracked back within 50 km of a deep convection event, and

orange dots are targets that were not.

**Figure 8.** GOES-16 0.64-μm visible channel imagery on 5 July 2018 at a) 2258 UTC, b) 2338

UTC, c) 2358 UTC, and d) 0023 UTC over central Arizona shown with every $20^{th}$ optical flow

vector in the x and y directions (subsampled for image clarity) illustrated with yellow wind barbs

(knots). Circles represent motion < 5 kts, which commonly occur over ground pixels.

**Figure 9.** The GOES-16 0.64-μm visible imagery shown with image targets backtracked from

subjective identification in Fig. 7a at 0023 UTC 6 July 2018 using the B04 method (blue/yellow)

and the Wu et al. (2016) approach (orange/red) at a) 0023 UTC, b) 2358 UTC, c) 2338 UTC and

d) 2213 UTC. Individual points are highlighted from each approach (yellow and red dots; see

text).

**Figure 10.** Color shaded wind speed for 0023 UTC 6 July 2018 over central Arizona shown

from a) the B04 optical flow method and b) the Wu et al. (2016) flow, shown with respective

flow vectors and the subjective position of the front edge of the OFB (blue line).



**14. Tables**
**Table 1.** Settings used in the Brox et al. (2004) successive over-relaxation scheme.

| Parameter | Value |
|---|---:|
| Outer Iterations (Pyramid Levels, nK) | 77 |
| Inner Iterations (nL) | 10 |
| Successive Over-Relaxation Iterations (nM) | 5 |
| Successive Over-Relaxation Parameter | 1.99 |
| Pyramid Scale Factor (SF) | 0.95 |
| $\gamma$ | 10 |
| $\alpha$ | 50 |


**15.  Figures**

**Step**
**1)**

Input Two Satellite Images
+Initial Flow Guess
$U \, \forall \, \Omega = [0\,,0]^T$

Do Outer Iterations K = 1, nK

**2)** Blur and Subsample Image with Scale Factor
$= SF^{(nK-K)}$, Scale $U$ to Image Pixel Size

**3)** Compute Warped Image Gradients
with $U$, Initialize $dU = [0\,,0]^T$      *B04 Eq. (8)*

Do Inner Iterations L = 1, nL

**4)** Compute Data and Smoothness
Terms with $U$ , $dU$      *B04 Eq. (10)*

Do Successive Over Relaxation Iterations M = 1, nM

**5)** Update $dU$ over
Subsampled Image      *Brox (2005) Section 4.2.3*

End Do M

End Do L

**6)** Update $U = U + dU$

End Do K

**7)** Navigate Resulting $\mathbf{x} + U$ to Convert to m s$^{-1}$


**Figure 1.** Flow chart of the B04 optical flow approach used here.  Note that SF, nK, nL and nM
are defined in Table 1.



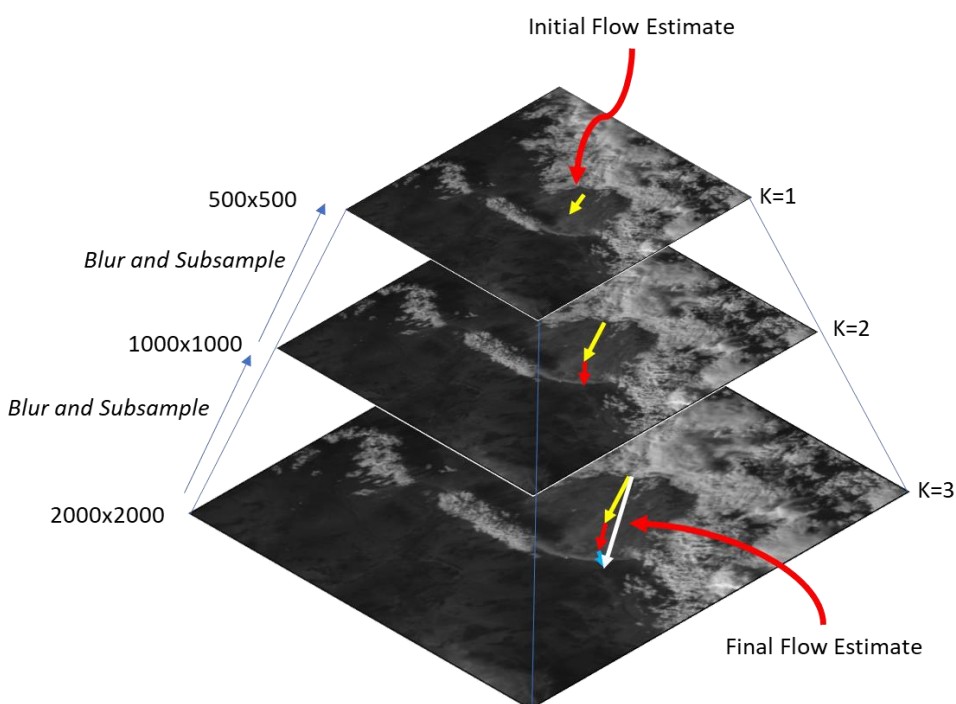


**Figure 2.** Schematic of coarse- to fine-scale warping optical flow in GOES-imagery. The largest displacements are found in the initial coarse grid (yellow arrow at the top of the pyramid), which are used as initial displacements for the next levels (red and blue arrows). The final displacement is the sum of each displacement estimate (white arrow). In this schematic, an example scale factor of 0.5 was used over 3 pyramid levels, in this work, a scale factor of 0.95 for 77 levels was used.

698



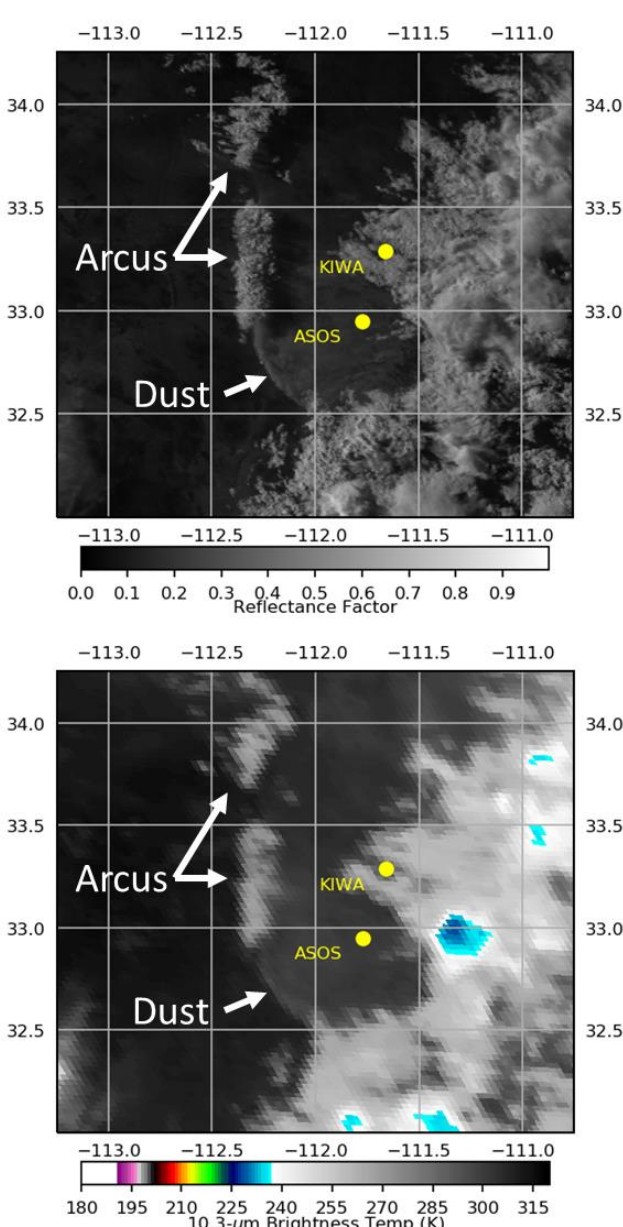

**Figure 3.** The 6 July 2018 0023 UTC GOES-16 0.64-μm visible reflectance (top) and $BT_{10.35}$
(bottom) over south-central AZ, centered on an OFB of interest.



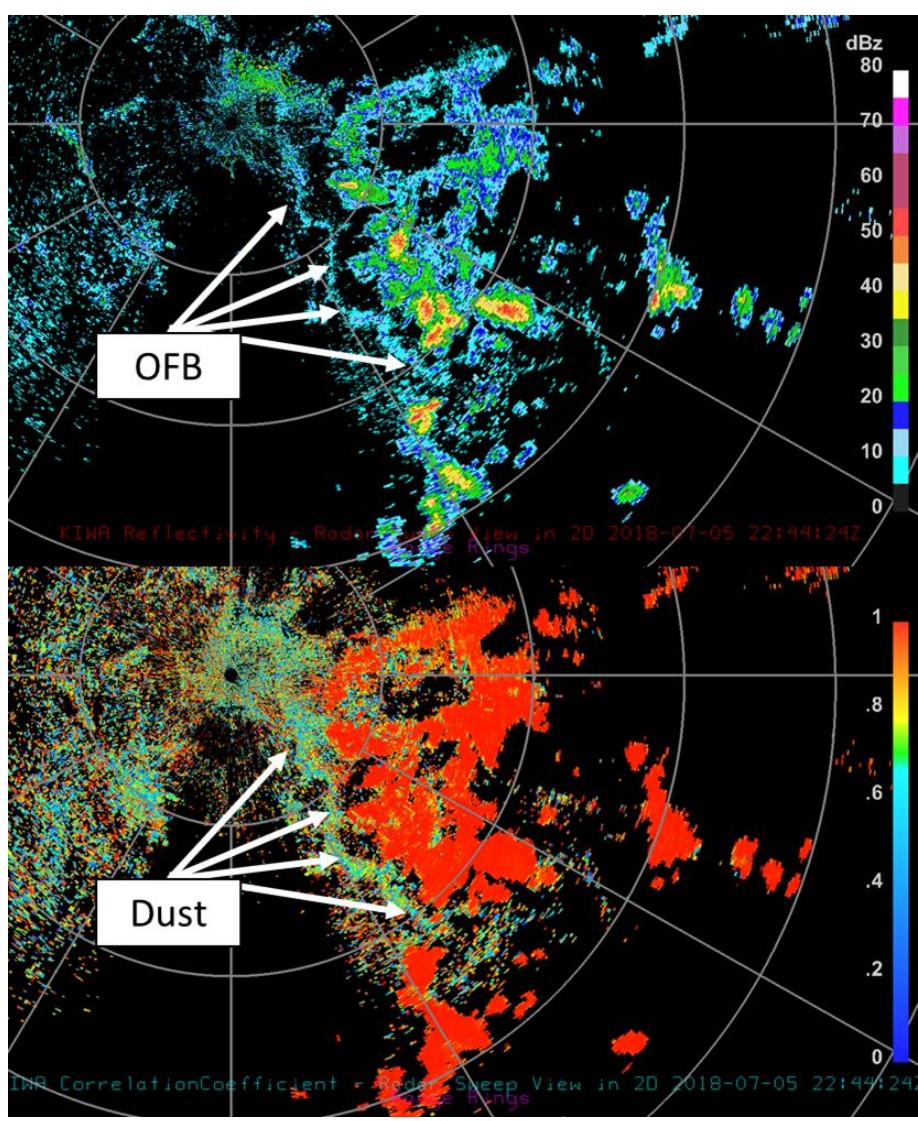

**Figure 4.** The KIWA Radar 2244 UTC 0.5° horizontal reflectivity (top) in dBZ and correlation coefficient (bottom). Range rings in grey indicate every 30° azimuth and 50 km in range.

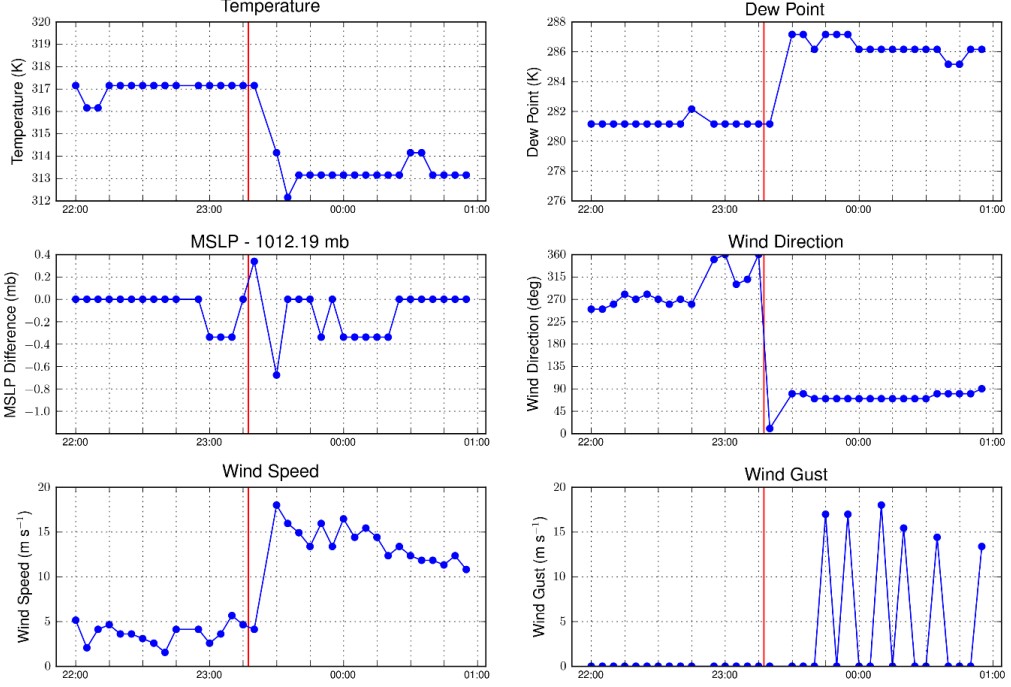


**Figure 5.** Surface High Frequency METAR observations of temperature (K; top left), dewpoint
(K; top right), mean sea level pressure (middle left), wind direction (° from N; middle right),
wind speed (m s⁻¹; bottom left), and wind gusts (m s⁻¹; bottom right). The surface station was
located at (32.95 °N -111.77 °E). The red line indicates the approximate time of boundary
passage over the station.




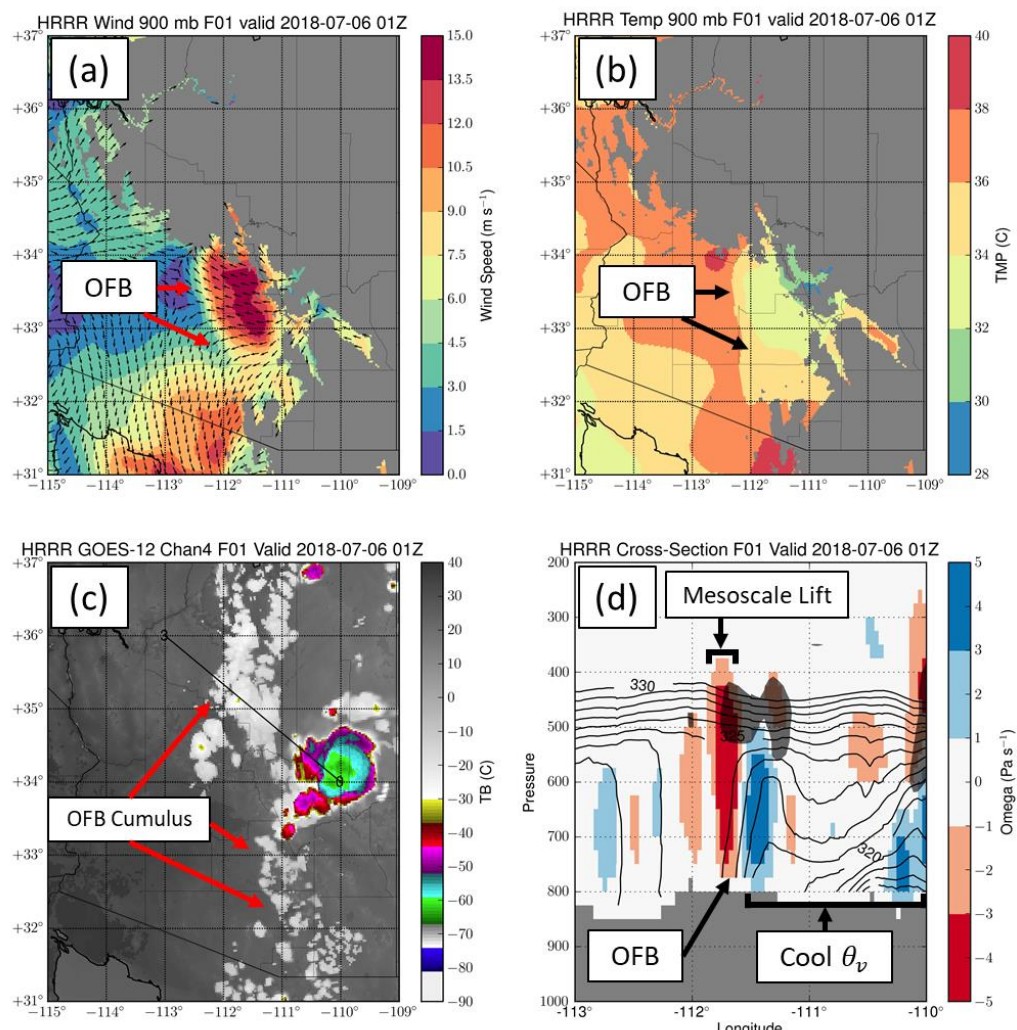


**Figure 6.** Four panel of HRRR output of OFB event, including a) wind speed, b) temperature, c)
simulated infrared brightness temperature, and d) a cross section along the black line in c with
virtual potential temperature $\theta_v$ in black contours (K), omega in color shaded pixels, and regions
of relative humidity > 90% highlighted with dark shading (bottom right).

719



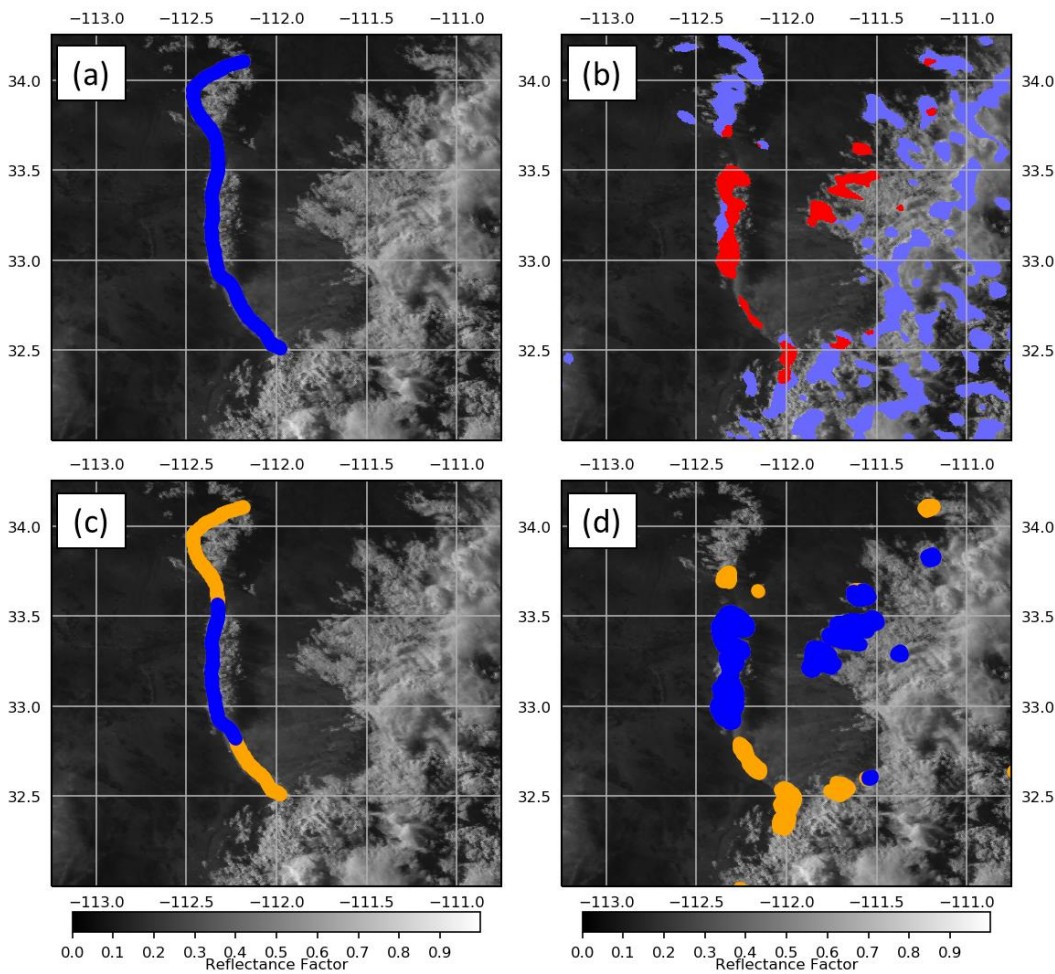

**Figure 7.** The 0023 UTC GOES-16 0.64-µm visible channel shown with a) subjectively
identified OFB (blue dots) and b) objectively identified linear features (blue shading). Also
shown are linear features that contained fast storm-relative motion (red shading). The results of
backtracking the c) subjectively and d) objectively identified OFB features are also shown,
where blue dots represent targets tracked back within 50 km of a deep convection event, and
orange dots are targets that were not.

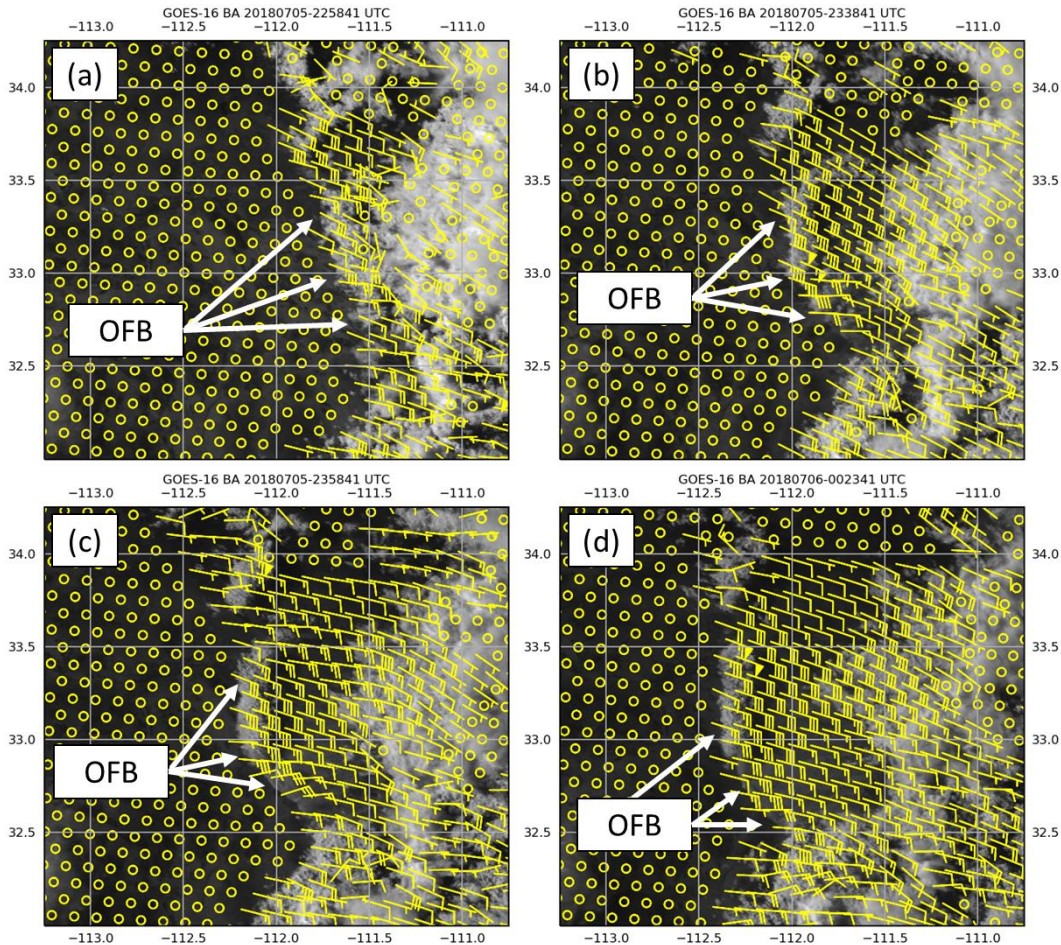

728

**Figure 8.** GOES-16 0.64-µm visible channel imagery on 5 July 2018 at a) 2258 UTC, b) 2338
UTC, c) 2358 UTC, and d) 0023 UTC over central Arizona shown with every 20th optical flow
vector in the x and y directions (subsampled for image clarity) illustrated with yellow wind barbs
(knots). Circles represent motion < 5 kts, which commonly occur over ground pixels.

733

734

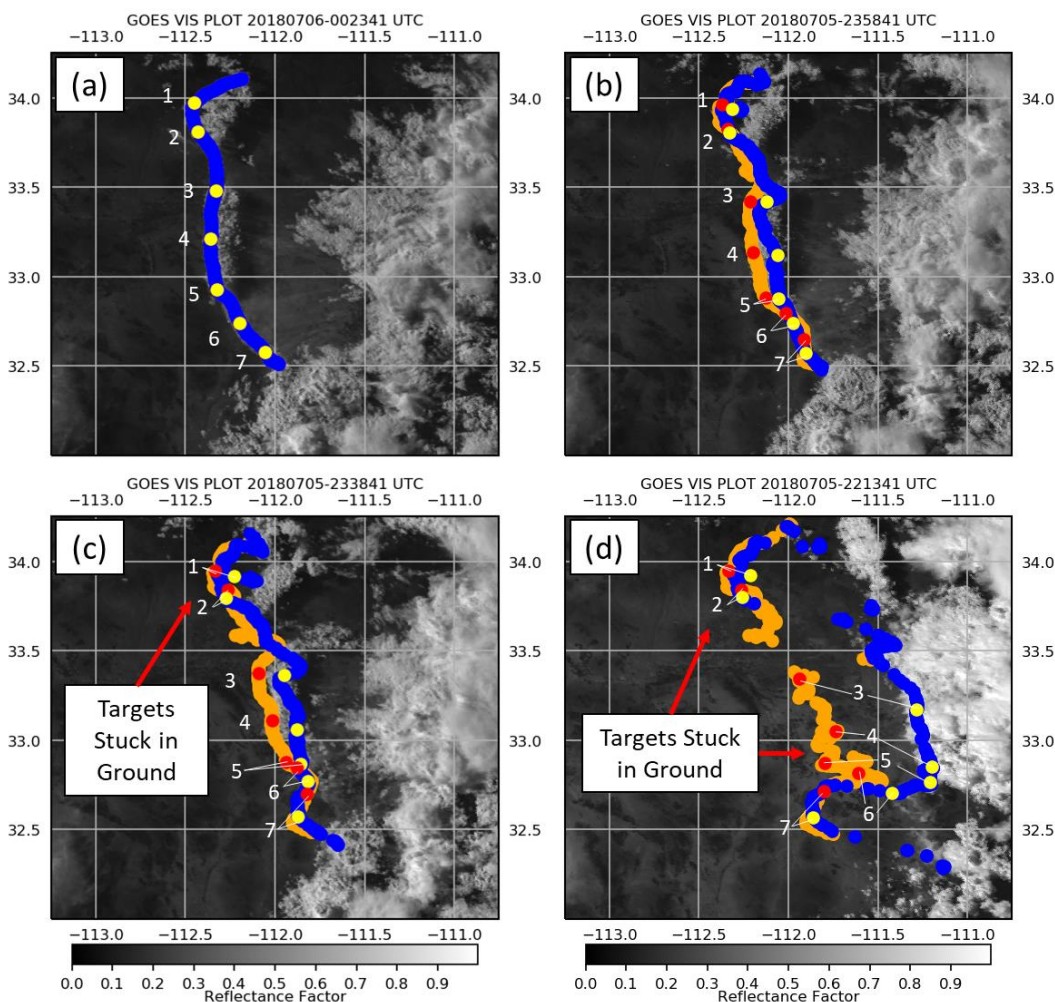

**Figure 9.** The GOES-16 0.64-μm visible imagery shown with image targets backtracked from subjective identification in Fig. 7a at 0023 UTC 6 July 2018 using the B04 method (blue/yellow) and the Wu et al. (2016) approach (orange/red) at a) 0023 UTC, b) 2358 UTC, c) 2338 UTC and d) 2213 UTC. Individual points are highlighted from each approach (yellow and red dots; see text).



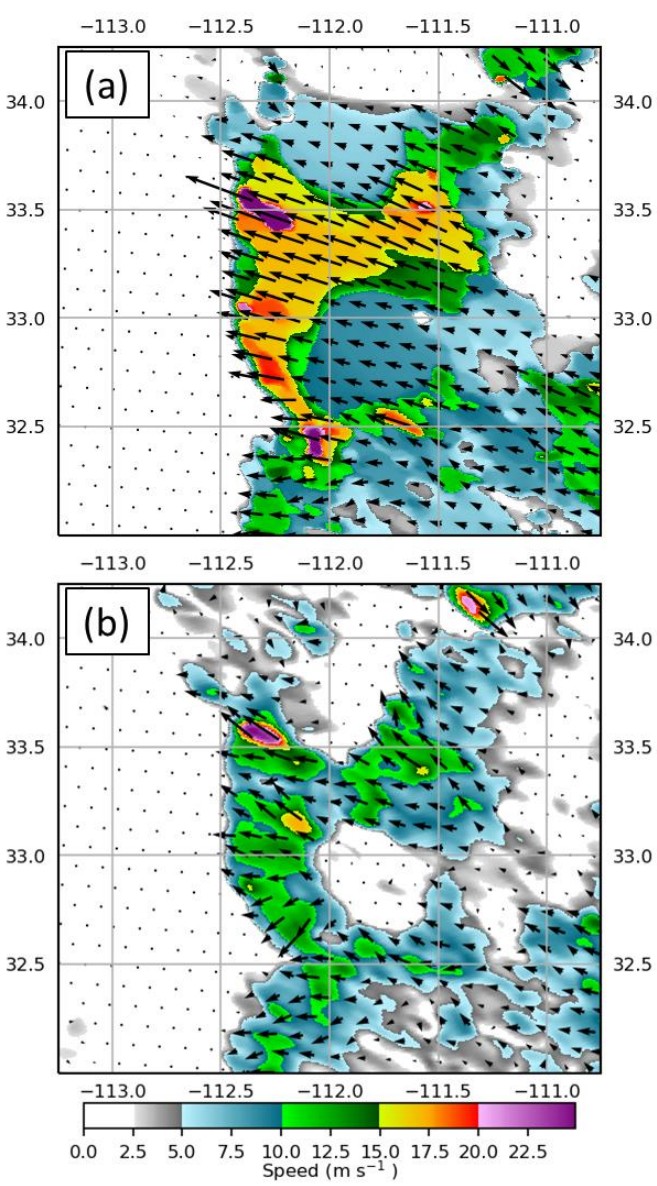

742

**Figure 10.** Color shaded wind speed for 0023 UTC 6 July 2018 over central Arizona shown
from a) the B04 optical flow method and b) the Wu et al. (2016) flow, shown with respective
flow vectors and the subjective position of the front edge of the OFB (blue line).