# Peer review of "Towards Objective Identification and Tracking of Convective Outflow Boundaries in Next-Generation Geostationary Satellite Imagery Jason M. Apke1, Kyle A. Hilburn1, Steven D. Miller1, and David A. Peterson2 1Cooperative Institute for Research in the Atmosphere (CIRA), Colorado State University, Fort Collins, CO, USA 2Naval Research Laboratory, Monterey CA, USA Corresponding Author: Jason Apke 3925A"

_Atmospheric Measurement Techniques, 2019_

## Referee Comment (RC1) · Anonymous Referee #2 · 14 Sep 2019

- General/Overall Comments

This work represents a unique application of an optical flow technique to high spatio-temporal geostationary imagery for the problem of identifying and tracking outflow boundaries. This paper is very well written and the work is well executed for the single case study described. Clearly, there is much more to be done and the author highlights these things, but I think this work is a great start and worthy of publication.

Recommend publication with only minor updates needed.

- Specific Comments

[Figure]

Line 10: change 15 min full disk to 10 min full disk (ABI Mode 6 is now used operationally)

Line 255: It is not clear to this reviewer how you define a "convective area". The word, "area", is used which would imply a group of pixels (presumably convective in this context). When working backwards toward this convection "area", is the convective area found when coming to the first pixel meeting the brightness temperature threshold?

- Technical Corrections

The following references cited in the text were not found in the References section:

Rotunno et al, 1988 Smalley et al 2007 Baker and Matthews, 2004 Van Den Broeke and Alsarraf, 2006

Line 165: Remove "spatially" from this sentence: "Thus,most optical flow computations initially spatially subsample images to where all displacements 166 are initially less than 1-pixel (Anandan, 1989; discussed more in Section 3.1), which can cause fast moving small features to be lost."

Line 285: replace ",etc." with ",for example".

―――――――――――――――――――――

---

## Author Comment (AC1) · 20 Sep 2019

"This work represents a unique application of an optical flow technique to high spatiotemporal geostationary imagery for the problem of identifying and tracking outflow boundaries. This paper is very well written and the work is well executed for the single case study described. Clearly, there is much more to be done and the author highlights these things, but I think this work is a great start and worthy of publication. Recommend publication with only minor updates needed."

We would like to thank the reviewer for the kind words and feedback provided on this article. Below is a detailed list of comments made and the authors' replies with details

on modifications made to the first submission of the papers.

"Line 10: change 15 min full disk to 10 min full disk (ABI Mode 6 is now used operationally)"

This has been fixed. (see Line 79)

"Line 255: It is not clear to this reviewer how you define a "convective area". The word, "area", is used which would imply a group of pixels (presumably convective in this context). When working backwards toward this convection "area", is the convective area found when coming to the first pixel meeting the brightness temperature threshold?"

In this paper, we define a convective area as any region within 50 km of a brightness temperature pixel $<$ -50 °C. This is not represented as a group of pixels here, rather a two-dimensional Haversine distance from all pixels below the brightness temperature threshold in the 10.3-$\mu$m image. This way, we can determine whether or not backward trajectories of targets we think are OFBs which are not forced to step along the pixel grid occur near deep convection. To alleviate reader confusion, we have reworded the sentence at Line 254 to: "If a back-traced pixel of the linear feature arrived within 50 km great-circle distance of a 10.35 $\mu$m brightness temperature (BT10.35) pixel lower than 223 K (-50 °C; using previous satellite imagery matched to the back-trajectory time), the original point was considered an OFB. The area subtended by the 50 km great circles derived from BT10.35 is hereafter referred to as the "deep convection area.""

"- Technical Corrections The following references cited in the text were not found in the References section: Rotunno et al, 1988 Smalley et al 2007 Baker and Matthews, 2004 Van Den Broeke and Alsarraf, 2006"

Good catch. A problem with the citation management software used has since been fixed and these references have been added to the reference section. To ensure this is not repeated on the next draft, all other in-text citations have been double checked for accompanying references.

"Line 165: Remove "spatially" from this sentence: "Thus,most optical flow computations initially spatially subsample images to where all displacements 166 are initially less than 1-pixel (Anandan, 1989; discussed more in Section 3.1), which can cause fast moving small features to be lost." "

Removed spatially from the sentence.

"Line 285: replace ",etc." with ",for example"."

For clarity, we decided to just remove "etc." all together, the sentence now reads:

"The combination of these model and observation datasets is employed to confirm the presence of a distinct convective OFB, rather than some other quasi-linear feature, such as a bore or elevated cloud layer." (Line 284)

———————————————————

---

## Referee Comment (RC2) · Sebastian Schmidt (Referee) · 14 Dec 2019

This manuscript compares different optical flow methods to backtrack suspected outflow boundaries (OFB) to the edge of the source deep convection cell - primarily a "local" vs. a "global" method. The unprecedented temporal and spatial resolution of current geostationary imagery seems to have attained the threshold where objective detection of such motion becomes feasible (although it is not stated what the threshold is). It should be noted that the study relies on traditional means of identification (weather stations and radar observations), and that the ultimate goal is to decrease the reliance of OFB on non-imagery ancillary data. At this point though, it seems that

the next objective would be the suppression of false positives, which arise from linear features in the convection itself (among others). This is mentioned by the authors in the study. They also emphasize that this may be the first step towards objective detection of these phenomena, but that much more work remains to be done. Despite these qualifications, it is impressive that semi-automated line detection with subsequent optical flow back-tracing "finds" the outflow boundary with such high fidelity (at least one of the methods), and the question is whether the best performing method always performs so well for different meteorological constellations. Also, as the authors mention, the remote sensing aspect of a future algorithm needs to be improved (by use of other spectral channels) to enhance contrast that will highlight linear features even for tell tale features that are less apparent than dust.

Comments: It was particularly hard to find reviewers for this paper, and in my capacity of associate editor for this paper, I therefore decided to provide this review as a substitute for a review, given the timeline of the process. I do not have any major comments except:

1) The description of the optical flow methods could be a bit more detailed and possibly be supported with graphs. Improving the manuscript in this regard is not a requirement, but in my opinion the somewhat dense text does prevent some readers from fully appreciating the manuscript, and why in the end one method "won out" over the other.

2) L247-250: These sentences are unclear. What is "calibrated to reflectance factor to isolate line features? First, "reflectance factor" should be clarified - is this simply reflectance in the native imagery? Second, what is calibrated to/by what, and how are line features actually isolated?

3) L299: Use of "low correlation coefficient" in the reflectivity to identify dust - can you briefly explain and/or provide a reference? This does not appear to be common knowledge.

4) L327: "Alternatively, storm-=relative motion from optical flow..." What is the motion

relative to - the convective core?

5) L382-386: This statement is a bit hard to follow. What is "background", for example? (I think I know, but it would be good stating this explicitly.)

6) Check that the grammar is correct - there are a few missing "the"s in a few places.

---

## Author Comment (AC2) · 25 Jan 2020

Key:

Reviewer's comments are in Red

*Authors' responses are in black italics.*

Comments: It was particularly hard to find reviewers for this paper, and in my capacity of associate editor for this paper, I therefore decided to provide this review as a substitute for a review, given the timeline of the process.

*Thank you for stepping in to review this paper, we appreciate the time and effort you have put in for us to get this manuscript published.*

I do not have any major comments except:

1) The description of the optical flow methods could be a bit more detailed and possibly be supported with graphs. Improving the manuscript in this regard is not a requirement, but in my opinion the somewhat dense text does prevent some readers from fully appreciating the manuscript, and why in the end one method "won out" over the other.

*Since so much progress has been made on the optical flow front without updates to satellite image motion tracking (e.g. AMVs), it is notably challenging to write a short and concise paper on the subject and connect it with the knowledge base of the typical meteorology and remote sensing researcher. In response to this comment, we have added a couple paragraphs to the background section 2.2 linking the current AMV optical flow approach (patch matching), which is a method most should understand, to the optical flow approach used here:*

*"Readers can contrast the HS method with the optical flow algorithm used in GOES AMVs, referred to as "patch matching" (PM; Fortun et al., 2015). In PM, a target (e.g. a 5x5 pixel box) identified as suitable for tracking is iteratively searched for in a sequential image within a reasonable search area (Fig. 1a). The motion is identified by which candidate target (e.g. another 5x5 pixel box displaced by the optical flow motion) in the sequential image best matches the initial target, typically by minimizing the sum-of-square error between the target and the candidate brightness values (Daniels et al., 2010; Nieman et al., 1997). The reader can draw similarities to the HS method by formulating the PM approach as an energy equation to be minimized,*

$$E(\boldsymbol{U}) = \sum_{n \in T} |I(\boldsymbol{x}_n, t) - I(\boldsymbol{x}_n + \boldsymbol{U}, t + \Delta t)|^2 \qquad (4)$$

*where the minimum in $E$ is found by computing Eq. (4) at every candidate target in the search region. As $E$ is only minimized within the target area $T$, PM represents a local method.*

*Research and extensive validation has shown that, with quality control, PM provides a valuable resource to derive and identify winds in satellite imagery (Velden and Bedka, 2009). However, there are several types of motions where PM would fail (Fig. 1b), many of which occur frequently in satellite OFB observations. AMVs found with Eq. (4) make two key assumptions, 1) that the brightness remains constant between sequential images at time $t$ and $t + \Delta t$, and 2) that the motion $\boldsymbol{U}$ is constant within the target. The first assumption, brightness constancy, fails when*

*there are excessive illumination changes in a sector that are not due to motion. These illumination changes may be due to evaporation or condensation, or simply due to changes in solar zenith angle throughout the day in visible imagery. The HS method also uses assumption 1), though it is relaxed when combined with the smoothness constraint. Assumption 2), which is not made in the HS method or other global methods, implies the PM method has no way to handle rotation, divergence, or deformation in an efficient manner, unless it is known apriori. Assumption 2) also fails to account for motion discontinuities, such as those near cloud-edges or within transparent motions. Furthermore, as there is no other constraint aside from constant brightness, PM methods struggle when there is little to no texture in the target and candidates. Quality control schemes are thus necessary to remove sectors that are poorly tracked with Eq. (4) in most AMV approaches.*

*PM was a popular method for AMVs over other optical flow approaches prior to the GOES-R era due to its simplicity, computational efficiency, and capability to handle displacements common in low-temporal resolution satellite imagery (Bresky and Daniels, 2006). …"* (==LINE 164-192==)

*We have also added a figure describing the patch matching approach, and a schematic of where it fails (see now Fig 1, at the end of this document for reference). This should clarify some of the nomenclature used on types of image motion.*

*We selected the Brox et al. (2004) optical flow approach due to it's simplicity, capability to handle regions patch matching could not (e.g. Fig. 1b), available open-source information (e.g. from opencv.org), and effective documentation (from the cited Brox et al., 2004 and Brox 2005 documents). We do not want to convey that it is the best optical flow method for the task at hand, as those techniques are evolving each year, so we have now clarified the methodology text with citations to the current validation datasets (a.k.a. benchmarks) used by the computer vision community:*

*"As recently overviewed in Fortun et al., (2015), there are several optical flow approaches that provide dense motion estimates which account for the weaknesses highlighted in Fig. 1b. Many have their own advantages and drawbacks in terms of computational efficiency, flexibility, and capability to handle large displacements, motion discontinuities, texture-less regions, and turbulent scenes. We selected an approach here by Brox et al. (2004) (Hereafter B04), given its simplicity, current availability of open-source information, and excellent documentation. The reader is cautioned, however, that dense optical flow is a rapidly evolving field, and research is currently underway to improve present techniques. While dense optical flow validation for satellite meteorological applications research like OFB identification is taking place, the reader is referred to the Middlebury (Baker et al., 2011), the MPI Sintel (Butler et al., 2012), and the KITTI (Geiger et al., 2012) benchmarks for extensive validation statistics of the most recent techniques using image sequences for more general applications."* (==LINE 229-240==)

*We feel a full validation of optical flow techniques for satellite motion tracking is beyond the scope of this research, though is almost certainly a topic of future work with the new capabilities*

*of current generation geostationary satellite imagers. With this manuscript, we simply want to show that accounting for local optical flow method deficiencies can help improve our capabilities to track and identify operationally relevant meteorological features.*

2) L247-250: These sentences are unclear. What is "calibrated to reflectance factor to isolate line features? First, "reflectance factor" should be clarified - is this simply reflectance in the native imagery? Second, what is calibrated to/by what, and how are line features actually isolated?

*This section has been revised for clarity. To answer your first question, the term "Reflectance Factor" was borrowed from the ABI Product Users Guide (Schmit et al., 2010), which is the radiance times the Kappa factor. While they are not the same thing, reflectance factor can be converted to reflectance by dividing by the cosine of the solar zenith angle. Calibrated was not the correct word to use here, so the statement has been revised. Line features are then isolated by convolving the provided filters with the reflectance factor and isolating where the resulting field is ≥ 0.02. To clarify this, we have added an additional equation step in the section. It now reads as follows:*

*"To handle the first step of line feature identification, a simple image line detection scheme was performed by convolving the original brightness field with a set of line detection kernels, so*

$$L = \sum_{i=1}^{4} a_i \star G(R) \qquad (9)$$

*where $\star$ is the convolution operator, $G$ is a gaussian smoothing function (using a 21x21 kernel and standard deviation of 5 pixels), R is the reflectance factor (radiance times the incident Lambertian-equivalent radiance, or the "kappa factor"; Schmit et al., 2010), L is the resulting line detection field, and $a_i$ represents the two-dimensional line detection kernels, defined as*

$$a_1 = \begin{bmatrix} -1 & -1 & -1 \\ 2 & 2 & 2 \\ -1 & -1 & -1 \end{bmatrix} a_2 = \begin{bmatrix} -1 & 2 & -1 \\ -1 & 2 & -1 \\ -1 & 2 & -1 \end{bmatrix} a_3 = \begin{bmatrix} 2 & -1 & -1 \\ -1 & 2 & -1 \\ -1 & -1 & 2 \end{bmatrix} a_4 = \begin{bmatrix} -1 & -1 & 2 \\ -1 & 2 & -1 \\ 2 & -1 & -1 \end{bmatrix}$$

*The resulting L field exhibits higher intensities where line features exist (Gonzalez and Woods, 2007). A threshold of $L \geq 0.02$ was used here to indicate a pixel contained a line feature. This method was compared to a subjective interpretation of boundary location for validation." (LINE 284-294)*

3) L299: Use of "low correlation coefficient" in the reflectivity to identify dust – can you briefly explain and/or provide a reference? This does not appear to be common knowledge.

*We have revised this sentence for clarity, and added citations to the relevant papers:*

*"The OFB was also captured in radar scans from KIWA at 2200 UTC (Fig. 4). The coincidence of low correlation coefficient (< ~0.5) and moderate to high reflectivity (near 20 dBZ) imply that the OFB contained non-meteorological scatterers (e.g. Zrnic and Ryzhkov, 1999). The radar measurements are consistent with previous reported values of lofted dust (Van Den Broeke and Alsarraf, 2016)." (LINE 343-347)*

4) L327: "Alternatively, storm-relative motion from optical flow..." What is the motion relative to - the convective core?

*We used the motion relative to the 0-6 km storm motion vector (which is a density weighted average of the layer flow) produced by the Global Forecast System numerical model here. The sentence has been reworded for clarity:*

*"Alternatively, the storm-relative motion (here > 15 m s$^{-1}$), or the motion relative to the 6 hr forecast field 0-6 km storm motion from the Global Forecast System (GFS) numerical weather prediction model run was used here to filter the false alarms (the red shading in Fig. 7b). The GFS forecast field was used over analysis to simulate what would be available globally in real-time." (LINE 373-376)*

5) L382-386: This statement is a bit hard to follow. What is "background", for example? (I think I know, but it would be good stating this explicitly.)

*We changed the ambiguous term "background" to "surface," and revised wording in this statement which should add clarity for readers. We were trying to state that convergence in the optical flow field only exists because there are stationary pixels ahead of the OFB. If this is the case, a faster OFB motion would then equal stronger convergence (so slower OFBs are less likely to be identified), which is undesirable for some types of products. We have revised this statement to:*

*"For this case study, it may have been possible to use convergence thresholding methods, analogous to radar-based objective OFB identification, to isolate the boundary. However, convergence as derived from the optical flow information here would only work because of local, stationary surface pixels ahead of the OFB. Thus, convergence would be stronger with faster OFB velocity, which is undesirable for an objective identification product as slow moving OFBs would be missed. The convergence would also be sensitive to nearby cloud structures ahead of the OFB which would exhibit different (non-stationary) motion from the surface." (LINE 427-434)*

6) Check that the grammar is correct - there are a few missing "the"s in a few places.

*We have checked the grammar and cleaned up the manuscript where necessary.*

[Figure]

**Figure 1.** Schematic of a) the PM optical flow scheme used by AMVs (e.g. Bresky et al., 2012), which finds a suitable target to track (e.g. the cloud at time 1), forecasts the displacement with numerical models (yellow arrow/dash box), and iteratively searches for the target at time 2 minimizing the sum-of-square error to get the AMV (red arrow), and b) example cloud evolution types mentioned in-text where the approach shown in (a) fails.